# Topping Inhibited Potassium Uptake via Regulating Potassium Flux and Channel Gene Expression in Tobacco

Taibo Liang [1,†], Huaxin Dai [1,†], Waleed Amjad Khan [2], Yadi Guo [1], Xiangyu Meng [3], Guiyao Wang [1] and Yanling Zhang [1,*]

1    Zhengzhou Tobacco Research Institute of CNTC, Zhengzhou 450001, China; liangtb@ztri.com.cn (T.L.); daihx@ztri.com.cn (H.D.); 07136@gxzy.cn (Y.G.); 11616080@zju.edu.cn (G.W.)
2    Tasmanian Institute of Agriculture, University of Tasmania, Hobart 7001, Australia; waleed.khan@utas.edu.au
3    Golden Leaf Manufacturing Center, China Tobacco Henan Industrial Co., Ltd., Zhengzhou 450016, China; xiangyumeng1992@163.com
*    Correspondence: zhangyanling@ztri.com.cn
†    These authors contributed equally to this work.

**Abstract:** Potassium ($K^+$) is mainly absorbed by plants from the soil and is primarily transported within the plant through the xylem. Topping has been reported to cause efflux and loss of $K^+$ in plants; however, its effect on the real-time flow rate and genotypes with varying $K^+$ accumulation ability is still unknown. Therefore, we carried out a pot experiment containing sand culture using two tobacco cultivars EY1 (high $K^+$ accumulating) and Y87 (low $K^+$ accumulating). The results demonstrated the change of $K^+$ flow direction from influx to efflux in the roots of both cultivars due to topping. The percentage ratio of $K^+$ efflux to influx was estimated to be 18.8% in EY-1 and 157.0% in Y87, respectively. We noticed a decline in indole acetic acid (IAA) content due to topping, which activated the expression of $K^+$ efflux channel gene *NTORK1* and inhibited the expression of $K^+$ influx channel genes *NKT1* and *NtKC1*. Furthermore, $K^+$ loss from the roots increased due to topping, which led to decreased $K^+$ concentration in entire tobacco plant. Topping had a more serious impact on the $K^+$ efflux rate and $K^+$ loss in Y87. IAA application after topping, in turn, decreased the $K^+$ loss in both the cultivars. We conclude that topping caused a decrease in IAA concentration and $K^+$ losses in tobacco leaves, and these losses can be mitigated by the exogenous application of IAA.

**Keywords:** topping; potassium; potassium flux; tobacco

## 1. Introduction

Topping is an agronomic approach commonly practiced during production of tobacco, cotton, and cannabis. It is performed by cutting off the main stem during the early reproductive stage of the plants to prolong the vegetative phase and achieve greater yields of quality leaves and fruits. However, topping may lead to a decrease in the potassium ($K^+$) content of tobacco leaves [1]. $K^+$ is one of the most essential nutrients in plants, and is critical for its growth and development [2,3]. It is also the most abundant ion in the plant cell [4], and participates in various physiological processes, including cell osmotic pressure balance, stomatal movement, enzyme activity, photosynthetic performance, assimilation product transport, and plant stress tolerance [5–8]. In tobacco, $K^+$ plays key roles in controlling important quality attributes of the leaves, e.g., aroma, combustibility, texture, color, and sugar and alkaloid content [9,10]. It has been demonstrated that the application of exogenous growth hormones (i.e., IAA, NAA) may help to reduce the $K^+$ losses in the plant due to topping and facilitates to increase the dry matter content of the tobacco leaves [11,12].

Potassium is mainly absorbed by the plant from soil through the roots and transported within the plant primarily through the xylem [13]. A theory was proposed by Ben-Zioni that corresponds to the significant role of $K^+$ in maintaining the translocation of $NO_3^-$ and

organic acid radical ions as an accompanying ion [14]. Thus, it helps to create a circulation flow balance between plant roots and shoots. It is of great interest to further elucidate the role of $K^+$ in the mechanisms underlying $K^+$ circulation within different plant parts of tobacco. The relationship between $K^+$ and calcium ($Ca^{2+}$) content in shoots and their flux changes in xylem sap has been considered one of the main factors in examining $K^+$ circulation patterns in *Lycopersicon esculentum* [15] and *Leptochloa fusca* [16]. Thus, low $K^+$ content in tobacco leaves is an economically undesirable constraint in tobacco, and could be resolved through in-depth understanding of ion flux changes and $K^+$ flow occurring in the plant roots.

Topping promotes the return of $K^+$ from the aerial parts to the roots [1]. This leads to increased loss of $K^+$ through the roots and reduced overall $K^+$ accumulation in the entire tobacco plant. It has been reported that the amount of returning $K^+$ from the shoots to the roots increases by 30% in response to topping [14,17]. Topping disturbs the expression of genes encoding $K^+$ channels in the roots, such as *NKT1*, *NtKC1*, *NTORK1*, and *NKT2* [1]. In the present study, we used two tobacco genotypes that differed with respect to their shoot $K^+$ accumulation ability. Therefore, they could be used to help us examine any relationship between shoot $K^+$ concentration and expression patterns of $K^+$ channel transporters due to topping.

In recent years, the non-invasive micro-test technique (NMT) has become a powerful tool for studying the dynamic changes of specific ions or molecules in biological samples [18–20]. The use of the NMT technique is beneficial for examining the changes produced in the movement of $K^+$ within the plant. In this study, we aimed to investigate the effects of topping and IAA treatment on the flux changes and the direction of $K^+$ flow in tobacco roots. These results are supported by revealing the expression of $K^+$ channel genes and deciphering the mechanism of $K^+$ regulation in tobacco.

## 2. Materials and Methods

### 2.1. Plant Material and Experimental Treatments

This experiment was conducted in the greenhouse facility of 'Zhengzhou Tobacco Research Institute' using two tobacco (*Nicotiana tabacum*) cultivars, 'EY-1' and 'Y87' as high- and low-potassium accumulation varieties, respectively [21]. A mixture of peat, perlite, and vermiculite (3:1:1) was used for germinating tobacco seedlings in trays. The germinated seedlings were then transferred to a seedbed to grow in a greenhouse with a daytime temperature of 25–30 °C, a nighttime temperature of 15–20 °C, and a relative humidity (RH) of 60–80%. The 50-day-old plants (with six fully emerged true leaves) were then transplanted to pots filled with quartz sand and continue to grow under the same glasshouse conditions. During the growth period, each pot was supplied with 200 mL of half-strength Hoagland's solution and refilled weekly to avoid any chances of drought.

There were two treatments in this experiment: topping (T1) and topping followed by IAA application (30 mg·kg$^{-1}$ concentration; T2), whereas intact plants without topping were used as controls (CK). Each treatment was grown in 30 pots. Topping was performed on the plants after 20-days of cultivation in quartz sand by cutting off the shoot apex with the help of sterilized scissors. A paste of 30 mg·kg$^{-1}$ IAA was immediately applied on the wounded part with a brush [22].

### 2.2. Determination of $K^+$ Concentration

The plant samples were harvested at regular intervals of 1 h, 6 h, 3 d, 6 d and 12 d after treatment application, with three plants being harvested at each time. The harvested plant samples were washed with distilled water first and then separated into stems and leaves, and roots. A subsample of fresh plant material was separated and stored under cold conditions until measurement of IAA content and gene expression. The rest was quenched at 105 °C for 15 min, and then oven-dried at 65 °C for 72 h, followed by grinding. Powdered plant samples were then digested with $HNO_3$-$H_2O_2$ with the help of microwave digester [23], and total $K^+$ concentration was measured using flame photometry [24].

### 2.3. Measurement of Root K+ Flux

Three tobacco plants were subjected to potassium starvation for 72 h prior to K+ flux measurements [25]. For this purpose, potassium nitrate and potassium dihydrogen phosphate were replaced with ammonium phosphate and ammonium nitrate in half-strength Hoagland's. Plants were taken into consideration for flux measurements after 3 days of K+ stress treatment. Two intact roots were selected from each plant to measure K+ flux rate at meristematic zone using NMT100 Series (Younger USA LLC, Amherst, MA, USA). The duration for each measurement was 300 s, with 6 technical replications for each treatment. The test liquid solutions used were as follows (mmol·L$^{-1}$): $KNO_3$ (0.1), $CaCl_2$ (0.1), MES (0.3), pH 6.0. Calibration solution I: $KNO_3$ (0.05), $CaCl_2$ (0.1), MES (0.3), pH 6.0. Calibration solution II: $KNO_3$ (0.5), $CaCl_2$ (0.1), MES (0.3), pH 6.0.

In addition, we selected six tobacco seedlings of uniform growth for each treatment to measure K+ loss. Two plant seedlings were immersed in a plastic box with 1.2 L starvation solution containing 0.2 mmol·L$^{-1}$ $CaSO_4$ and 5 mmol·L$^{-1}$ MES (pH: 5.7). We used three boxes for each treatment. Starvation solution (0.5 mL) was collected for the K+ concentration test at 0 h, 3 h, 6 h, 12 h, 24 h, 36 h, 48 h, 60 h, 72 h, 96 h and 120 h. The K+ concentration in starvation solution reflects the K+ loss in plants.

### 2.4. Measurement of Hormone IAA Concentration

A subsample (0.1 g) of fresh leaf sample was ground to a powder using pestle and mortar by adding liquid nitrogen. Pre-cooled 80% methanol (1 mL) was added into the powdered sample and then left in the dark at 4 °C for 24 h. The extracts were centrifuged at $6000 \times g$ at 4 °C for 10 min, and the supernatant was collected. The resultant sample was extracted once again with the addition of 0.5 mL 80% methanol. Nitrogen blowing instrument (Hangzhou Mio Instrument Co., Ltd., Hangzhou, China) was used to blow the samples to approximately 0.5 mL. Then, 0.5 mL of petroleum ether was added into the mixture and shaken well. The supernatant was removed carefully by keeping the lower aqueous phase solution and this procedure was repeated thrice. Nitrogen blowing device was operated again to blow the lower aqueous phase near to dryness, and the mobile phase was used to make the final volume up to 0.5 mL. The resultant solution was poured into a sample bottle with the help of syringe filters.

IAA concentration was determined using Rigol L3000 high-performance liquid chromatography instrument (Rigol Technology Co., Ltd., Suzhou, China) according to a method described by Meulebroek et al. [26]. Kromasil C18 reversed-phase chromatographic column (250 mm × 4.6 mm, 5 μm) was used in this analysis and the hormone standard of analytical grade (HPLC) was purchased from SIGMA. The mobile phase was consisted of methanol: 1% acetic acid water (2:3, *v/v*) with the injection volume of 10 μL. The fluxes were set at 0.8 mL·min$^{-1}$, with the column temperature and the detection wavelength being adjusted at 35 °C and 254 nm, respectively.

### 2.5. Expression Analysis of K+ Channel Genes

Fresh leaf and root tissue samples were collected and frozen immediately in liquid nitrogen, followed by storage at −80 °C. Total RNA was extracted from the plant samples using TaKaRa MiniBEST Universal RNA Extraction Kit (Takara, Tokyo, Japan). The quality and concentration of extracted RNA samples were assessed using Nanodrop 2000 spectrophotometer (Thermo Fisher Scientific, Waltham, MA, USA). These RNA samples were then reverse transcribed into cDNA using the QuantiTect Reverse Transcription Kit (Qiagen, Hilden, Germany).

Full-length cDNA sequences of putative K+ channel genes (e.g., *NKT1*, *NtKC1*, and *NTORK1*) were downloaded from NCBI database. *ACTIN* was used as an internal reference gene. The primer sequences of genes used in the present study were accessed from previous studies [27], as shown in Table 1.

**Table 1.** Primer sequences of K$^+$ channel genes in tobacco used for analyzing gene expression.

| Gene | Transcript ID | Primer Sequence (5′–3′) |
|:---:|:---:|:---:|
| *NKT1* | *AB196790* | F: GGCTCGTCTAACGGCAGATT<br>R: CAAGCACAACCCTTCCACCT |
| *NtKC1* | *AB196791* | F: CACTATTGTCATGGCGGATG<br>R: TCTTCGGTACATCCGTTTCTG |
| *NTORK1* | *AB196792* | F: AGTGAAACAACTTGAGAGTACCTC<br>R: GAGAAGCATAAACTGCTACAGTGG |
| *ACTIN* | *AB158612* | F: AACAGTTTGGTTGGAGTTCTGG<br>R: CATGAAGATTAAAGGCGGAGTG |

All gene expression analysis was performed using a LightCycler 480 II fluorescent quantitative real-time PCR instrument (Roche, Mannheim, Germany). The SYBR Premix Ex Taq™ PCR Reagent kit was purchased from Takara Bio Inc. (Otsu, Japan). Each PCR reaction consisted of 2 μL cDNA, 2 μL (1 + 1, forward and reverse) primer sequence, 10 μL SYBR Green, and 6 μL ddH$_2$O, with the final volume of 20 μL. The PCR reaction conditions were set as follows: pre-denaturation at 95 °C for 5 min, denaturation at 95 °C for 10 s, annealing at 55 °C for 15 s and extension at 72 °C for 15 s, with 45 total cycles. The gene expression analysis was performed using LightCycler®480 software (version 1.5.1.62, Mannheim, Germany), and the threshold value (Ct) of the genes was obtained. The Ct value of *ACTIN* was subtracted to obtain the ΔCT value and $2^{-\Delta\Delta CT}$ was evaluated to estimate the relative expression of the genes [27].

*2.6. Statistical Analysis*

One-way analysis of variance (ANOVA) was performed by following the least significant difference (LSD) test in SPSS (v19.0) software. All the graphs were plotted using GraphPad Prism 8.0. *p*-values < 0.05 were considered as statistically significant in this study throughout.

**3. Results**

*3.1. Effect of Topping on the K$^+$ Concentration of Tobacco*

Topping had a significant negative effect on the K$^+$ accumulation in shoots of both tobacco varieties (Figure 1). The shoot K$^+$ concentration in EY-1 decreased by 12.17% and 10.39% at 6 d and 12 d after topping is performed, respectively. For Y87, the shoot K$^+$ concentration decreased by 6.53% and 7.58% at 6 d and 12 d. The K$^+$ concentration in shoots of EY-1 increased by 6.63% and 7.82% at 6 d and 12 d, respectively, under IAA treatment in comparison with topping. While the shoot K$^+$ concentration in Y87 after IAA application increased only by 1.29% and 2.23% at 6 d and 12 d, respectively. These results showed that the topping reduced the shoot K$^+$ accumulation in tobacco and the extent of decrease may varied between genotypes. Moreover, the effect of IAA application after topping was also found to be cultivar-specific and only observed in cultivar 'EY-1'.

*3.2. Effects of Topping on Root K$^+$ Flux of Tobacco*

We found significant differences in root K$^+$ flux between the two varieties under control conditions (Figure 2). The root K$^+$ influx rate of cultivar EY-1 was 1.72 times greater than that of Y87. Topping led to the change of K$^+$ flow direction from influx to efflux. To visualize the effect of topping on K$^+$ flux, the ratio of net K$^+$ efflux after topping: net K$^+$ influx in the control was estimated in both the cultivars. The percentage ratio of K$^+$ efflux to influx was 18.8% and 157.0% in EY-1 and Y87, respectively. The K$^+$ flux increased by 60.75 pmol·cm$^{-2}$·s$^{-1}$ in EY-1 and 47.00 pmol·cm$^{-2}$·s$^{-1}$ in Y87 with topping as compared to the control. K$^+$ efflux in Y87 was 2.98-times greater than that of EY-1 in response to topping. The K$^+$ efflux rate decreased by 85.96% in EY-1 and 60.30% in Y87 after the application of IAA as compared to topping. Therefore, it is obvious that IAA treatment after topping

has been demonstrated to be effective in reducing the $K^+$ efflux from the roots, while maintaining the $K^+$ flow within the plant tissues.

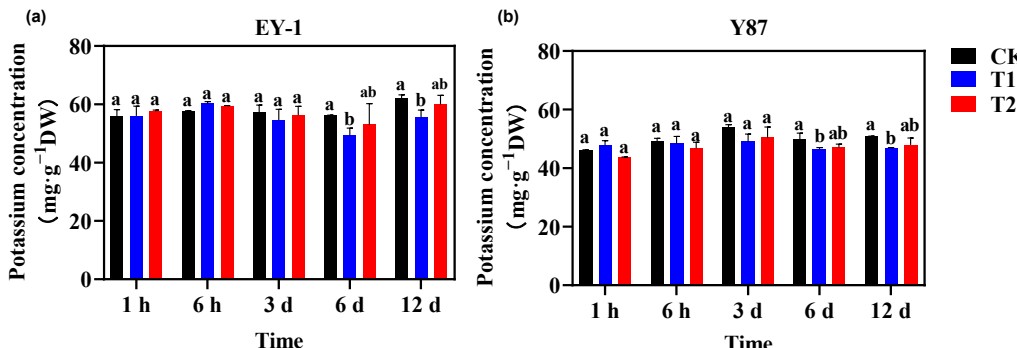

**Figure 1.** Changes of $K^+$ concentration in shoots of EY-1 (**a**) and Y87 (**b**) at 1 h, 6 h, 3 d, 6 d, and 12 d after topping. CK: control, T1: topping, T2: IAA application after topping. Each value represents the mean of three biological replicates ± SD value. Different letters indicate significant difference among treatments at $p < 0.05$.

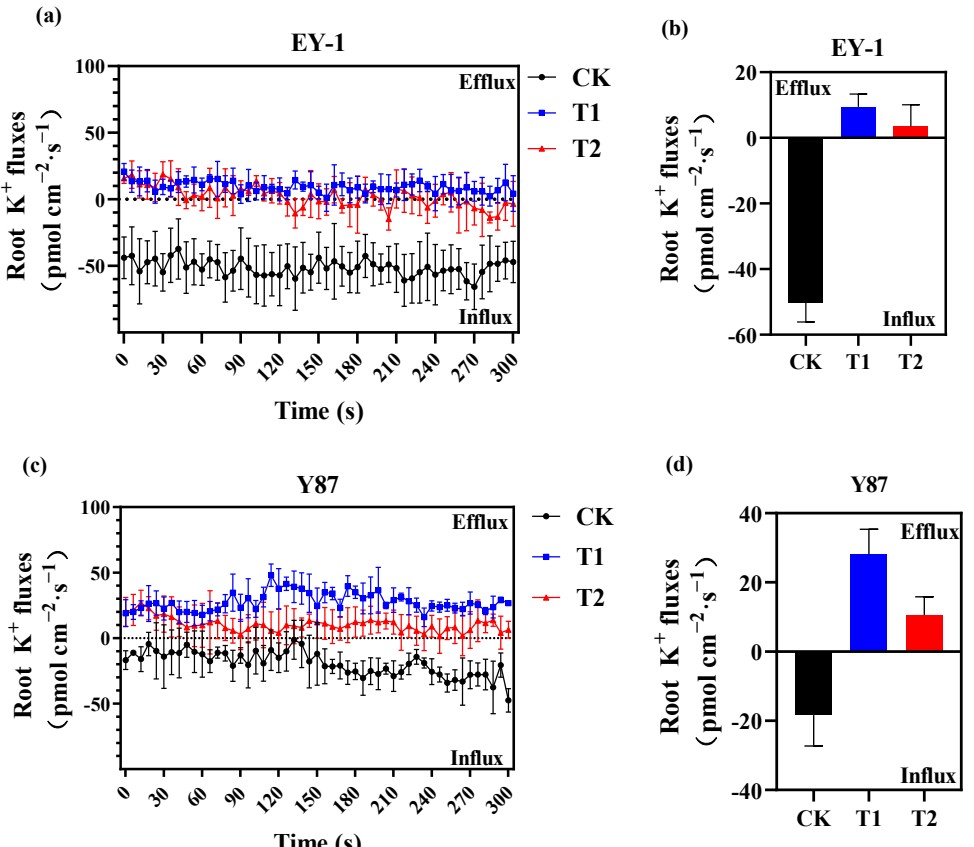

**Figure 2.** The root $K^+$ instantaneous flux of EY-1 (**a**) and Y87 (**c**), and average flux of EY-1 (**b**) and Y87 (**d**) measured in roots. CK, control; T1, topping; T2, IAA application after topping. Each seedling was tested for 300 s, and each treatment had 6 biological replicates. The change of $K^+$ flux between the control and the treatment was calculated as mean flux.

### 3.3. Effects of Topping on $K^+$ Loss of Tobacco

The $K^+$ concentration in the starvation solution was highest within 6 h of treatment, and then decreased gradually with the absorption of $K^+$ by the roots (Figure 3). The $K^+$ concentrations in the starvation solution of EY-1 topping treatment were 35.75% and 17.03% higher than that of the control at 3 h and 6 h, respectively, whereas such $K^+$ concentrations

were 46.05% and 28.94% higher in Y87. $K^+$ loss of Y87 caused by topping was greater than that of EY-1 (Figure 2). IAA application decreased the $K^+$ concentration in starvation solution by 13.82% and 7.20% as compared to T1 treatment in EY-1 at 3 h and 6 h, respectively, whereas the decrease in the $K^+$ concentration of Y87 in the starvation solution were 3.71% and 1.61% lower in T2 than T1. Thus, these results showed that IAA application followed by topping alleviated the $K^+$ loss in plants caused by topping.

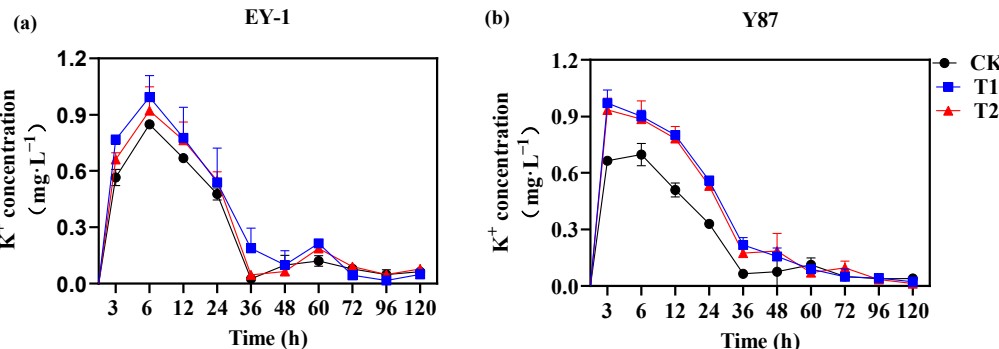

**Figure 3.** $K^+$ concentration in starvation solution for EY-1 (**a**) and Y87 (**b**). CK, control; T1, topping; T2, IAA application after topping.

### 3.4. Effects of Topping on Tobacco Endogenous Hormone IAA Content

The synthesis of IAA mainly occurs in the flourishing part of the tobacco plant, which can be inhibited by topping [17]. The IAA content was observed to be consistent in both the varieties under control conditions (Figure 4). The IAA content in both varieties decreased immediately due to topping but increased after 6 h. The initial decrease in IAA content could be due to the sudden removal of the apical dominance in plants, and this damage was gradually recovered after 6 days with the emergence of axillary buds. IAA application significantly increased the IAA content in shoots by 3.78% and 12.94% averagely in EY-1 and Y87 as compared to topping. These results suggested that the decrease in IAA content in plant shoots due to topping could be mitigated by IAA application.

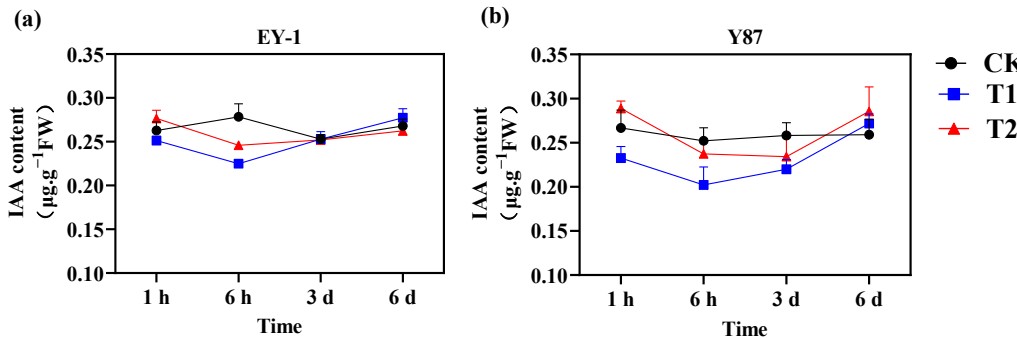

**Figure 4.** Changes of endogenous hormone IAA content in shoots of EY-1 (**a**) and Y87 (**b**). CK, control; T1, topping; T2, IAA application after topping. The mean $\pm$ SE of three replicate samples are shown as error bars. FW: fresh weight.

### 3.5. Effects of Topping on the Expression of $K^+$ Channel Genes in Tobacco

#### 3.5.1. Expression Analysis of $K^+$ Efflux Channel Gene *NTORK1*

The gene *NTORK1* is expressed in guard cells of leaves and parenchyma cells of root epidermis, and mainly participates in long-distance transport of $K^+$. The expression of $K^+$ efflux channel gene *NTORK1* in leaves and roots were observed to be different between the two tobacco varieties in response to topping (Figure 5). The gene was significantly up-regulated at 1 h, 6 d and 12 d in EY-1 leaves compared to control, while it showed opposite trend in Y87 leaves. Topping significantly increased the *NTORK1* expression in Y87 roots, while no significant effect was found in EY-1 roots (except at 3 d). It was observed

that the up regulation of *NTORK1* in Y87 root after topping may be related to root $K^+$ loss, and the response of *NTORK1* in EY-1 root was weak.

IAA application after topping decreased the *NTORK1* expression in leaves of EY-1 at 1 h and 12 d, but the effect on roots was not consistent. IAA application after topping decreased the expression of *NTORK1* both in leaves and roots in Y87 at most time intervals compared to with the topping treatment. It can be concluded that topping combined with IAA could induce the down-regulation of gene *NTORK1* expression in tobacco leaves, especially in Y87, which plays a key role in inhibiting and alleviating $K^+$ efflux caused by topping.

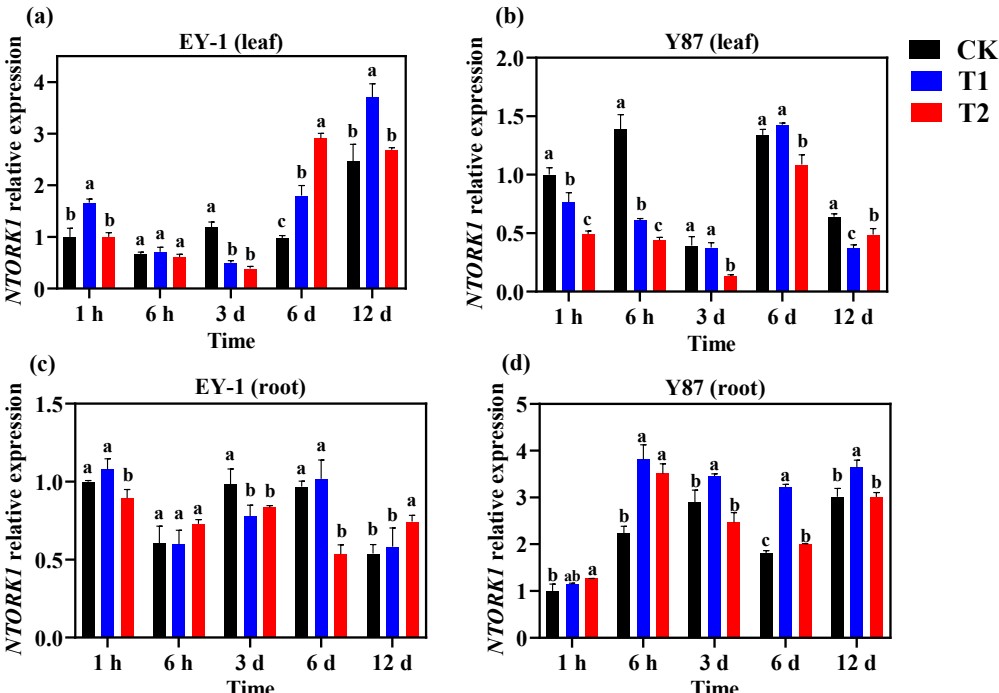

**Figure 5.** Relative expression levels of the *NTORK1* gene in EY-1 leaf (**a**), root (**c**) and Y87 leaf (**b**), root (**d**) as determined by real-time PCR. CK, control; T1, topping; T2, IAA application after topping. The mean ± SE of three replicate samples are shown as error bars. Means with same lowercase letters have no significant difference ($p$ = 0.05) in the LSD test.

### 3.5.2. Expression Analysis of $K^+$ Influx Channel Gene *NKT1*

The expression of $K^+$ influx channel gene *NKT1* was significantly decreased in EY-1 leaves after topping, while there was no consistent effect in Y87 leaves (Figure 6). The *NKT1* expression in Y87 root was inhibited significantly by topping at most time intervals. Thus, the response of *NKT1* expression to topping varied between the two varieties, and the response of leaves and roots of the same variety to topping was also different. Topping inhibited the *NKT1* expression in the leaves of high $K^+$ concentration cultivar EY-1.

IAA application increased the expression of *NKT1* in EY-1 leaves at 1 h, and Y87 leaves at 6 h and 12 d, while the *NKT1* expression of Y87 roots at 3 d and 12 d were increased after topping. The differences in the *NKT1* expression of EY-1 in roots between the control and topping plants were not significant during most sampling periods.

### 3.5.3. Expression Analysis of $K^+$ Influx Channel Gene *NtKC1*

Topping had a major effect on the expression of $K^+$ influx channel gene *NtKC1* in the leaves of the two varieties (Figure 7). The expression of *NtKC1* gene in EY-1 leaves decreased at 6 h and 12 d, while in roots it was decreased at intervals of 1 h, 6 h and 3 d after topping. For the low $K^+$ cultivar Y87, the expression of *NtKC1* gene in roots was decreased at most sampling times after topping, while the expression in leaves was found inconsistent.

The response of gene *NtKC1* to IAA treatment varies with different tobacco varieties and organs. IAA application increased the expression of *NtKC1* gene in EY-1 leaves at 1 h, while increased *NtKC1* gene expression in root at 6 h and 3 d after topping. For the Y87, the expression of *NtKC1* gene was decreased in both leaves and roots due to topping at most sampling times.

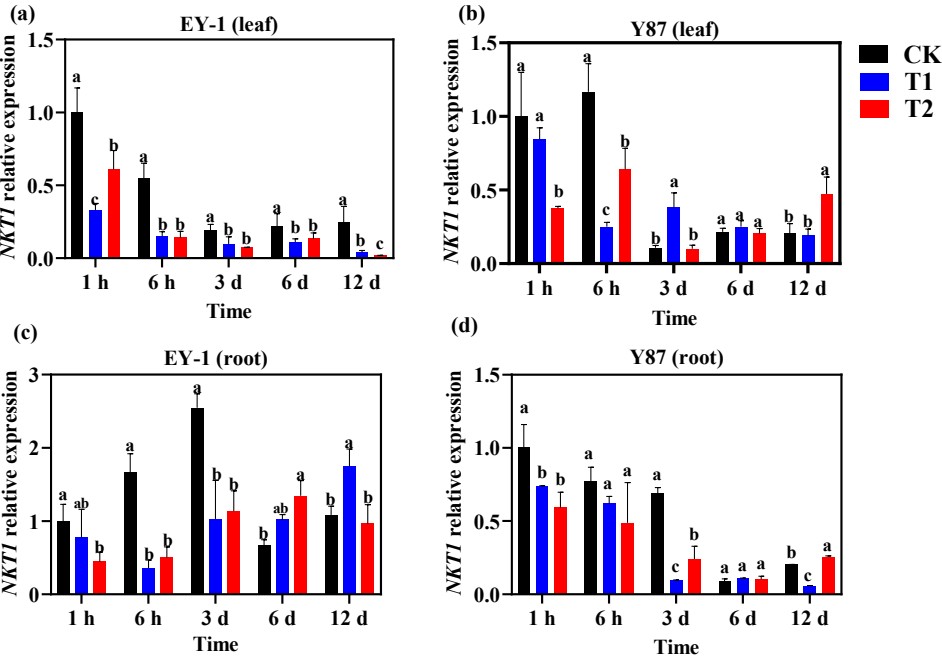

**Figure 6.** Relative expression levels of the *NKT1* gene in EY-1 leaf (**a**), root (**c**) and Y87 leaf (**b**), root (**d**) as determined by real-time PCR. CK: control, T1: topping, T2: IAA application after topping. The mean ± SE of three replicates are shown as error bars. Means with same lowercase letters have no significant difference ($p = 0.05$) in the LSD test.

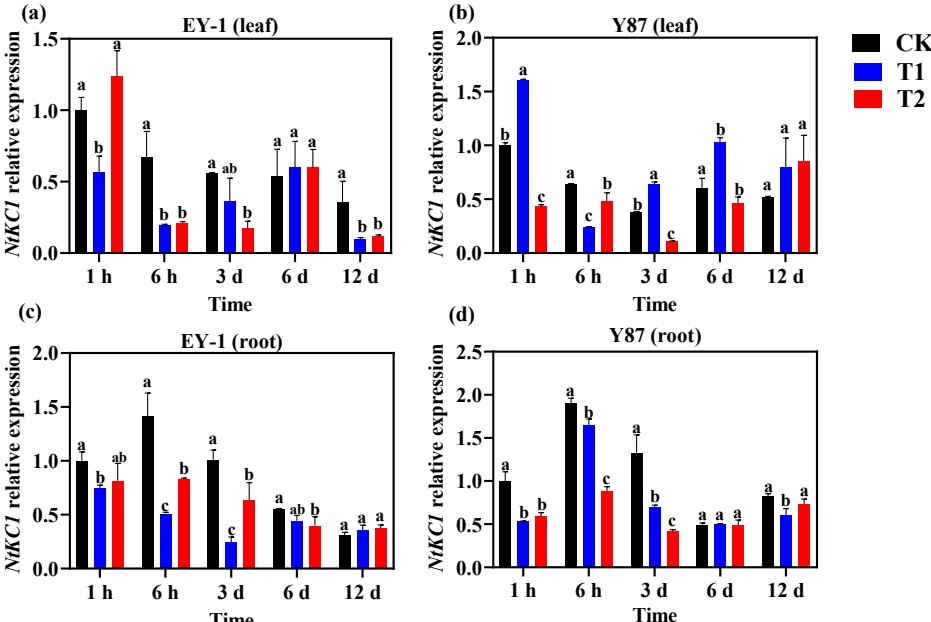

**Figure 7.** Relative expression levels of the *NtKC1* gene in EY-1 leaf (**a**), root (**c**) and Y87 leaf (**b**), root (**d**) as determined by real-time-PCR. CK, control; T1, topping; T2, IAA application after topping. The mean ± SE of three replicate samples are shown as error bars. Means with same lowercase letters have no significant difference ($p = 0.05$) in the LSD test.

## 4. Discussion

It has been demonstrated that K$^+$ is phloem-mobile and flows from shoot to root in response to topping [28,29]. Topping has been reported to cause efflux and loss of K$^+$ in plants; however, its effects on genotypes are still unclear. NMT is widely used to precisely measure the absorption of certain ions in the plant roots [30]. The NMT test revealed that the real-time K$^+$ velocity in tobacco roots varies with cultivars of different K$^+$ accumulation ability. The root K$^+$ influx rate was found to be greater in the high K$^+$ accumulating cultivar than in the low K$^+$ cultivar, which was an important reason for the difference of K$^+$ accumulation. Topping leads to the K$^+$ loss in tobacco plants, which was proven by the detection of change of K$^+$ flux direction through NMT and starvation test. The change in K$^+$ velocity in tobacco roots caused by topping is an important reason for the decline of K$^+$ concentration in tobacco plants. There were also genotypic differences in the effect of topping on K$^+$ flow rate. Topping induced higher ratio of K$^+$ efflux to influx in Y87, indicated that the K$^+$ loss of low K$^+$ cultivar caused by topping was more serious.

The absorption and transport of K$^+$ in plants are regulated by K$^+$ channel genes. *NTORK1* is an efflux K$^+$ channel gene found in tobacco, which has 65–75% homology with *SKOR* and *GORK* of *Arabidopsis* [31]. In this study, the *NTORK1* expression was up-regulated after topping, especially for low K$^+$ accumulation tobacco cultivar Y87. This result was consistent with the conclusion of Dai. et al. [1] in tobacco. Topping significantly increased the *NTORK1* expression in Y87 roots, which might be related to root K$^+$ loss in plants. *NKT1* is a K$^+$ influx channel gene, which mainly exists in the plasma membrane of cells. It is activated and opened under the condition of membrane hyperpolarization voltage, and participates in the regulation of K$^+$ absorption [32,33]. *NtKC1* is another K$^+$ influx channel gene obtained from tobacco using homologous cloning technology. It is mainly distributed in root hairs and endothelium, and plays a regulatory role in the absorption of K$^+$ from the soil by tobacco. In this study, topping inhibited the expression of *NKT1* and *NtKC1* in roots of low K$^+$ accumulation cultivar Y87. However, the down-regulation of *NKT1* and *NtKC1* genes due to topping mainly occurred in leaves. The differences in organ gene response to topping in different varieties might be involved in causing K$^+$ loss. Topping regulated the expression of K$^+$ channel genes, which caused increased K$^+$ efflux in the roots, ultimately leading to reduced K$^+$ concentration in the entire tobacco plant.

Studies have shown that the uptake of K$^+$ by plants is related to the exchange of H$^+$ and K$^+$ promoted by auxin; that is, auxin could stimulate H$^+$ secretion while absorbing K$^+$ at the same time [34,35]. Auxin could promote the root primordium of tobacco roots to grow and promote rooting, thereby enhancing the absorption of K$^+$ [36–38]. Topping could destroy the synthesis and transport of apical auxin, thus affecting the absorption of K$^+$ by tobacco roots [39]. In this study, the auxin and K$^+$ contents of two tobacco varieties were inhibited by topping. The losses of IAA content due to topping was gradually eliminated with the growth of axillary buds in plants. Exogenous IAA application proved to be effective in alleviating losses in IAA and K$^+$ content in tobacco plant. As a signal substance, IAA could regulate gene expression, and then promote plant root growth and K$^+$ absorption [40]. This study found that IAA significantly inhibited the expression of outer channel gene *NTORK1* in Y87, which plays an important role in preventing K$^+$ loss. However, the effect of exogenous IAA on K$^+$ channel genes varied in the different K$^+$ accumulating varieties, thus necessitating further research for in-depth understanding of K$^+$ regulation mechanism in tobacco.

## 5. Conclusions

Topping caused the change of K$^+$ flow direction in tobacco roots, which led to increased K$^+$ loss, and decreased K$^+$ concentration in entire tobacco plants. This was mainly due to the effect of topping, which inhibited IAA synthesis by removing the apical buds of plants, activating the K$^+$ outflow channel gene *NTORK1* and inhibiting the expression of K$^+$ influx channel genes. There were significant genotypic differences in the harmful effect of topping

on K$^+$ flow in tobacco plants. The low K$^+$ accumulating cultivar was more susceptible to K$^+$ loss than the high K$^+$ accumulating one due to topping. The exogenous IAA could be effective in mitigating the losses in shoot K$^+$ concentration due to topping in tobacco.

**Author Contributions:** Conceptualization, T.L. and H.D.; methodology, X.M.; software, Y.G.; validation, T.L., H.D. and Y.Z.; formal analysis, G.W.; investigation, X.M.; resources, X.M.; data curation, X.M.; writing—original draft preparation, Y.G.; writing—review and editing, T.L., H.D. and W.A.K.; visualization, W.A.K.; supervision, Y.Z.; project administration, Y.Z.; funding acquisition, Y.Z. All authors have read and agreed to the published version of the manuscript.

**Funding:** This research was funded by Zhengzhou Tobacco Research Institute Dean Fund (112017CA0100) and key project of China National Tobacco Corporation (11202102037).

**Institutional Review Board Statement:** Not applicable.

**Informed Consent Statement:** Not applicable.

**Data Availability Statement:** Not applicable.

**Conflicts of Interest:** The authors declare no conflict of interest.

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
