# Peer review of "Topping Inhibited Potassium Uptake via Regulating Potassium Flux and Channel Gene Expression in Tobacco"

_agronomy, doi:10.3390/agronomy12051166_

Round 1

Reviewer 1 Report

Topping is a necessary measure in tobacco cultivation and production, and which has a direct impact on the maturity of tobacco leaves. Studying the changes of substances in tobacco after topping is of great significance for tobacco production. In this study, authors analyzed the efflux and loss of K+ in high K+ tobacco cultivar (EY1) and low K+ tobacco cultivar (Y87). The results showed that topping made the K+ flow direction in the roots changed from influx to efflux. In addition, IAA application decreased the K+ loss of the two tobacco varieties. The exogenous application of IAA has an important compensatory effect on the negative effects of topping. This manuscript can be published in IJMS after minor revision. Some comments are provided below regarding some suggestions about how the manuscript might be improved.

Point 1: Full text symbols should be in Times New Roman format. Authors need to review and revise the full text symbols, such as ℃ in Line79, 95, 96 ……

Point 2: The data in Figures 1 and 2 lack significance analysis.

Point 3: In fig. 2, the effect caused by the IAA application looks very minor, it would be great to present the results of a gradient e.g. 10, 20, 30, 40, and 50 mg·kg-1.

Point 4: Some words in the manuscript need italics. “p” in Line 183, “Arabidopsis thaliana” in Line 426 ……

Point 5: In fig 6, relative expression levels of the NKT1 gene in control should be used as standard, and then compared the expression level between different tissues.

Point 6: Substantial language editing is required in some sections of the manuscript.

Author Response

(1) Full text symbols should be in Times New Roman format. Authors need to review and revise the full text symbols, such as ℃ in Line79, 95, 96 ……

Response: We used Palatino Linotype plaint text font in this MS, which is uniform throughout according to template from the journal.

(2) The data in Figures 1 and 2 lack significance analysis.

Response: Figure 1 already has significance analysis, we added significance analysis in Figure 2.

(3) In fig. 2, the effect caused by the IAA application looks very minor, it would be great to present the results of a gradient e.g. 10, 20, 30, 40, and 50 mg·kg-1.

Response: In our results, the potassium efflux rate decreased by 85.96% (EY-1) and 60.30% (Y87) by the application of IAA as compared to topping, which (we consider) indicates a significant difference between the two different treatments. As for the reviewer suggestion of using more concentration of IAA, we used only one concentration in this experiment, which was adopted according to Dai et al. (2008). However, we acknowledge the fact that using two or more concentrations would be essential in better understanding of the relationship between K+ efflux and IAA treatment.

(4) Some words in the manuscript need italics. “p” in Line 183, “Arabidopsis thaliana” in Line 426 ……

Response: It has been revised according to the comments of the reviewer.

(5) In fig 6, relative expression levels of the NKT1 gene in control should be used as standard, and then compared the expression level between different tissues.

Response: We revised the Fig 6 according to the reviewer’s comments. The relative expression level of the NKT1 gene at 1 h in control was used as the standard. We have made similar changes in Fig 5 (NTORK1) and Fig 7 (NtKC1).

(6)Substantial language editing is required in some sections of the manuscript.

Response: We revised the language of the manuscript throughout.

Reviewer 2 Report

K+ plays key roles in controlling important quality attributes of the leaves e.g., aroma, combustibility, texture, color, sugar, and alkaloid content. They aimed to investigate the effects of topping and IAA treatment on the flux changes and the direction of K+ flow in tobacco roots. They found results revealing the expression of K+ channel genes and deciphered the K+ regulation mechanism in tobacco. And they determined that exogenous IAA could compensate for the negative effects caused by topping. Topping is one of the most important processes in broadleaf tobacco production, so the results are important. The introduction provides sufficient background and includes all relevant references. The research was designed appropriate, the methods adequately were described, and the conclusion was supported by the results. The research is original and has the potential to attract readers' interest.

Author Response

No comments

Reviewer 3 Report

Minor corrections:

  1. Materials and methods: Line 106-109: There is much repetition of units need to simplify it. Can write units in the start of sentence in brackets followed by the name of chemical/compound etc  and its strength. In the present form can be confusing for readers. It would be better if the treatments (line 106 - 114)are shown in the form of table.
  2. Results:
  3. There are certain formatting and typo errors in the manuscript. For example a missing space before bracket (in text as well as title of figures, for example Figure 5). When figure is mentioned in the text its written as Fig 1 or Fig.1. Authors need to keep the same format while mentioning Figures. For reference see line 203 - 213.
  4. If possible can create one figure for comparing data of both varieties EY-1 and Y87, in figure 1.
  5. Figure 6 and 7: space is missing in title

Author Response

(1) Materials and methods: Line 106-109: There is much repetition of units need to simplify it. Can write units in the start of sentence in brackets followed by the name of chemical/compound etc  and its strength. In the present form can be confusing for readers. It would be better if the treatments (line 106 - 114)are shown in the form of table.

Response: We revised the units according to the comments. Because this involves two different small experiments, it is not represented in the form of a table.

(2) Results:There are certain formatting and typo errors in the manuscript. For example a missing space before bracket (in text as well as title of figures, for example Figure 5). When figure is mentioned in the text its written as Fig 1 or Fig.1. Authors need to keep the same format while mentioning Figures. For reference see line 203 - 213.

Response: We fixed the typo and format errors according to the reviewer’s suggestions.

(3)If possible can create one figure for comparing data of both varieties EY-1 and Y87, in figure 1.

Response: We think we can find the difference between the two varieties from Figure 1, so we don't need to add another figure.

(4)Figure 6 and 7: space is missing in title

Response: The space was added in title of Figure 6 and 7.

Reviewer 4 Report

It was known that topping may lead to decrease in potassium (K+) uptake and K+ content in tobacco leaves. In this paper, the authors provided some evidences from K+ flux measured by NMT and expression of several genes related to K+ uptake. However, there were some obvious problems in the manuscript as below.

My major concerns include the details in material and methods as well as analysis of results.

Authors said that each treatment has 15 replicates, but we do not know how many pots or plants for each replicate. Also, we do not know how many pots or plants were sampled for most traits they tested. For example, determination of K+ concentration in 2.2, measurement of root K+ flux in 2.3, measurement of IAA in 2.4, expression of K+ channel genes and transport genes in 2.5. In addition, authors need to explain why they determined K+ at 1 h, 6 h, 3 d, 6 d and 12 d. According to the description, the K+ flux were measured in roots of big plants (50+20 days old), we do not know what roots (isolated or intact roots) and what part of roots were tested. As for the determination of K+ efflux in starvation solution, it was not clear how plants were assigned. One plant per box, or six plants per box? More importantly, authors did not tell us why they choose NKT1, NtKC1, NtHAK1 and NTORK1 to do gene expression analysis. In general, the aa sequences of K+ channels and transporters in tobacco and Arabidopsis should be collected to generate the phylogenetic tree, and then choose the close homologs of Arabidopsis. Lastly, AtKC1 negatively regulates the AKT1-mediated K+ uptake in Arabidopsis roots, i.e. inhibited the AKT1-mediated inward K+ currents (Wang et al., Cell Research (2010) 20:826-837). However, NtKC1 seems to be regarded as an inward-rectifying K+ channel in this paper.

In terms of Results section, there are several analysis that are inconsistent with Figures. They were listed following.

Line 172-174: In fact, both varieties did not show significant differences between topping and topping with IAA application.

Line 221: 6-days should be 6 h.

Line 236: 6 d, not 6 h.

Line 238: Actually, there was a significant difference at 3 d.

Line 260: 12 d, not 6 d.

Besides these major concerns, some other problems were annotated in manuscript.

Author Response

(1) My major concerns include the details in material and methods as well as analysis of results.

Authors said that each treatment has 15 replicates, but we do not know how many pots or plants for each replicate. Also, we do not know how many pots or plants were sampled for most traits they tested. For example, determination of K+ concentration in 2.2, measurement of root K+ flux in 2.3, measurement of IAA in 2.4, expression of K+ channel genes and transport genes in 2.5.

Response: We revised this section to clarify our methods. We used three plants each time to determine the potassium content, IAA content and gene expression. Three plants were used for the measurement of root K+ flux, and another six plants were used for the K+ loss determination.

(2)In addition, authors need to explain why they determined K+ at 1 h, 6 h, 3 d, 6 d and 12 d. According to the description, the K+ flux were measured in roots of big plants (50+20 days old), we do not know what roots (isolated or intact roots) and what part of roots were tested.

Response: At present, the time and amount of potassium loss caused by topping are not clear. According to previous studies, the gene response after topping is faster, which may change within a few hours. The hormone response is also relatively rapid, and will change within a few hours or 1 day. The response of plant potassium content is relatively slow, which may not be shown until a few days later. Therefore, considering the changes of several indicators, we determined K+, IAA and gene expression at 1 h, 6 h, 3 d, 6 d and 12 d.

In fact, we measured the meristem of the root tip. And it has described in detail in this text.

(3)As for the determination of K+ efflux in starvation solution, it was not clear how plants were assigned. One plant per box, or six plants per box?

Response: We explained in detail in the manuscript according to the comments of the reviewer. In fact, Two plant seedlings were put in a plastic box with 1.2 L starvation solution containing 0.2 mmol·L-1 CaSO4 and 5 mmol·L-1 MES (pH: 5.7), three boxes per treatment.

(4)More importantly, authors did not tell us why they choose NKT1, NtKC1, NtHAK1 and NTORK1 to do gene expression analysis. In general, the aa sequences of K+ channels and transporters in tobacco and Arabidopsis should be collected to generate the phylogenetic tree, and then choose the close homologs of Arabidopsis. Lastly, AtKC1 negatively regulates the AKT1-mediated K+ uptake in Arabidopsis roots, i.e. inhibited the AKT1-mediated inward K+ currents (Wang et al., Cell Research (2010) 20:826-837). However, NtKC1 seems to be regarded as an inward-rectifying K+ channel in this paper.

Response: So far NKT1, NKT2, NtKC1 and NTORK1 were cloned from BY-2 cells of tobacco. Among them, NKT1, NKT2, NtKC1 were recognized as inward rectifier, whereas NTORK1 was outward rectifier (Sano et al., 2007; Dai et al., 2008). Those genes have been widely studied to evaluate the effect of potassium absorption in tobacco, so we chose them as our target genes. In line with their affiliation to the known subfamilies of plant Shaker-like K+ channels, NKT1, NKT2, and NtKC1 resembles to the Arabidopsis thaliana AKT1, AKT2/3, and AtKC1 subfamilies, respectively, while NTORK1 is homologous to members of the A. thaliana SKOR/GORK subfamily (Kasukabe et al., 2006; Sano et al., 2007). We also noticed that AtKC1 inhibited the AKT1-mediated inward K+ currents and negatively shifted the voltage dependence of AKT1 channels (Wang et al., Cell Research(2010)20:826-837). However, there are studies where NtKC1 was also suggested as a potassium inward rectifier gene (Chen et al. J Nanobiotechnol (2020) 18:21; Da et al., 2009; Sano et al., 2009). Therefore, based on the previous research conclusions in tobacco, we believe that NtKC1 is an inward rectifier gene.

(5)In terms of Results section, there are several analysis that are inconsistent with Figures. They were listed following. Line 172-174: In fact, both varieties did not show significant differences between topping and topping with IAA application.

Response: We agree with reviewer that there are no major differences between topping and IAA application after topping, but minor differences are still present between two varieties. So, we have revised this section accordingly.

(6)Line 221: 6-days should be 6 h.

Line 236: 6 d, not 6 h.

Line 238: Actually, there was a significant difference at 3 d.

Line 260: 12 d, not 6 d.

Response: We revised them according to the comments of the reviewers

(7)Besides these major concerns, some other problems were annotated in manuscript.

Response: We carefully checked the problems marked by the reviewer and made corresponding modifications. The details can be seen in the revised manuscript.